# Co-Expression Analysis of microRNAs and Proteins in Brain of Alzheimer’s Disease Patients

**DOI:** 10.3390/cells11010163

**Published:** 2022-01-04

**Authors:** Callum N. Watson, Ghazala Begum, Emma Ashman, Daniella Thorn, Kamal M. Yakoub, Moustafa Al Hariri, Ali Nehme, Stefania Mondello, Firas Kobeissy, Antonio Belli, Valentina Di Pietro

**Affiliations:** 1Institute of Inflammation and Ageing, University of Birmingham, Edgbaston, Birmingham B15 2TT, UK; cal97@live.co.uk (C.N.W.); ghazalabegum@hotmail.com (G.B.); eka894@alumni.bham.ac.uk (E.A.); DXT882@student.bham.ac.uk (D.T.); k.yakoub@bham.ac.uk (K.M.Y.); ali.hassan.nehme@gmail.com (A.N.); a.belli@bham.ac.uk (A.B.); 2Surgical Reconstruction and Microbiology Research Centre, National Institute for Health Research, Queen Elizabeth Hospital, Birmingham B15 2TH, UK; 3Department of Emergency Medicine, American University of Beirut Medical Center, Beirut 1107 2020, Lebanon; ma147@aub.edu.lb; 4Department of Biomedical and Dental Sciences and Morphofunctional Imaging, University of Messina, 98165 Messina, Italy; stm_mondello@hotmail.com; 5Program for Neurotrauma, Neuroproteomics & Biomarkers Research, Departments of Emergency Medicine, Psychiatry, Neuroscience and Chemistry, University of Florida, Gainesville, FL 32611, USA; firasko@gmail.com

**Keywords:** Alzheimer’s disease, microRNA profile, protein profile, co-expression analysis, Rap1 signalling

## Abstract

Alzheimer’s disease (AD) is the most common form of dementia globally; however, the aetiology of AD remains elusive hindering the development of effective therapeutics. MicroRNAs (miRNAs) are regulators of gene expression and have been of growing interest in recent studies in many pathologies including AD not only for their use as biomarkers but also for their implications in the therapeutic field. In this study, miRNA and protein profiles were obtained from brain tissues of different stage (Braak III-IV and Braak V-VI) of AD patients and compared to matched controls. The aim of the study was to identify in the late stage of AD, the key dysregulated pathways that may contribute to pathogenesis and then to evaluate whether any of these pathways could be detected in the early phase of AD, opening new opportunity for early treatment that could stop or delay the pathology. Six common pathways were found regulated by miRNAs and proteins in the late stage of AD, with one of them (Rap1 signalling) activated since the early phase. MiRNAs and proteins were also compared to explore an inverse trend of expression which could lead to the identification of new therapeutic targets. These results suggest that specific miRNA changes could represent molecular fingerprint of neurodegenerative processes and potential therapeutic targets for early intervention.

## 1. Introduction

Alzheimer’s disease (AD) is the most common cause of neurodegenerative dementia, leading to severe disability and early death, and with devastating consequences for families, health-care systems, and society [1]. It affects approximately between 40 and 50 million globally, but the number of cases is expected to triple by 2050 owing to population growth and ageing [2,3]. Most AD cases are considered sporadic, and the cause is unclear; however, a notable risk factor is carrying the *APOE4* allele of the Apolipoprotein E gene [4]. Fewer than 1% of AD cases are known to be familial, caused by inheritance of rare autosomal dominant mutations [5] of amyloid precursor protein (*APP*) gene, presenilin 1 (*PSEN1*), and presenilin 2 (*PSEN2*), where *PSEN1* and *PSEN2* are involved in the processing of APP to amyloid-beta (Aβ) [6].

Diagnosis of Alzheimer’s disease is based on a history of cognitive decline combined with neurological and laboratory analysis, including the measurement of cerebrospinal fluid (CSF) biomarkers of Aβ and tau tangles and neuroimaging scans to identify brain atrophy and anatomical changes [7]. The histological hallmarks of AD pathology are Aβ-plaques and tau tangles accompanied by neuron loss in the brain. The pathology follows a specific pattern of brain regions in which plaques and tangles are found and mapped according to Braak staging [8].

Although the FDA recently approved the first Alzheimer’s medication clearing out amyloid plaques, this intervention is still controversial, and to date, no treatment has shown a clear/undoubted patient clinical benefit. This may be a consequence of the late-stage diagnosis of patients as well as the fact that the numerous studied compounds are unable to modify the disease process [9]. Therefore, there is an urgent need to discover early mechanistic biomarkers/endpoints as well as new therapeutic targets.

MicroRNAs (miRNAs, miRs), a small non-coding RNA class of post-transcriptional regulators, are particularly attractive molecules due to their ability to regulate multiple genes. Technology has now been developed to isolate, quantify, and profile miRNAs in diseases and is being combined with approaches to deliver miRNA therapeutics effectively without an immunogenic response [10,11,12,13,14,15,16,17]. Whilst these developments are promising, there are currently no miRNA therapeutics available; however, there is a remarkable effort in place, with many entering clinical trials with the earliest implication for cancer [18]. Consequently, miRNA as a therapy presents a promising solution for multifactorial diseases, such as AD, which do not yet have successful disease-modifying treatments; however, further research is required to elucidate the potential role of miRNA in this complex pathology. Previous studies in AD patients have highlighted changes in the expression of miRNAs that target AD-related mRNA and proteins. Among them, miR-101 and miR-16 have been shown to target APP mRNA [19,20]. MiR-124 was downregulated in the AD model SAMP8 mice, leading to enhanced expression of the splicing regulator PTBP1 and aberrant APP splicing [21]. Several miRNAs are involved in the amyloid cascade through the regulation of beta-secretase 1 (BACE1), including miR-107, miR-29a/b-1, miR-29c, miR-188-3p, miR-339-5p, miR-195, miR-186, and miR-124 in animal and human studies [22,23,24,25,26,27,28]. Other miRNAs that target the catalytic subunits of γ-secretase have also been shown to be deregulated in AD. Furthermore, Tau protein, the main constituent of neurofibrillary tangles, has been shown to be regulated by miR-132 and miR-34a [29].

In this study, we present an integrated analysis of miRNA and protein expression from the brains of AD patients. The primary outcome was to elucidate the pathological mechanism that can lead to neuronal death in confirmed and late AD cases. This was reached through the expression analyses of miRNAs and proteins and their related pathways in controls and AD (Braak V-VI stage) patients. The secondary outcome had the aim to investigate whether any of these pathological mechanisms or pathways detected in the late stage of AD were also present at the early stage of AD (Braak III-IV) and consequently whether any of the molecules differentially expressed can be used as early indicators of the pathology or as potential targets to stop or delay the progress of the pathology.

## 2. Materials and Methods

### 2.1. Sample Population

Post-mortem temporal cortex brain tissues were used in this study, which had been taken within 72 h of death and obtained from the South-West Dementia Brain Bank (SWDBB, Bristol, UK) and stored at −80 °C. Samples were stratified by Braak tangle stages I-VI. Stages I and II were used as the control group (*n* = 5), stages III-IV were used for the early stage of AD group (*n* = 5), and stages V and VI were used for the late-stage of AD group (*n* = 5). The groups were sex- and age-matched to remove these variables as a potential confounding factor.

### 2.2. MicroRNA Analysis

#### 2.2.1. RNA Extraction and Analysis

Temporal cortex samples were homogenised for 30 s using a TissueRuptor II (Qiagen, Hilden, Germany) in RNase-free PBS (ThermoFisher Scientific, Waltham, MA, USA). RNA was extracted from the homogenised samples using the miRNeasy Mini Kit (Qiagen, Hilden, Germany) according to the manufacturer’s protocol. The RNA concentration was measured on a nanophotometer (IMPLEN, Westlake Village, CA, USA).

An Agilent 2100 Bioanalyzer (Santa Clara, CA, USA) was used to detect the size distribution of total RNA as well as determine the quality of the RNA. A total of 100 ng of RNA was extracted and converted to cDNA using the miRCURY LNA kit (Qiagen, Hilden, Germany). cDNA was mixed with a SYBR master-mix (miRCURY SYBR Green Master Mix; Qiagen, Hilden, Germany). This was added to the miRCURY LNA miRNA miRNome Panel I–II using the automated pipettor the QIAgility (Qiagen, Hilden, Germany). These plates were run on the Quantstudio 5 qPCR machine (ThermoFisher Scientific, Waltham, MA, USA) as instructed by protocol provided with the panels. In total 751miRNAs (miRCURY LNA miRNA miRNome PCR Panels, https://geneglobe.qiagen.com/us/product-groups/mircury-lna-mirna-mirnome-pcr-panels, accessed on 6 October 2019 were tested using the miRNome plates for each of the samples. Overall, 705 miRNAs showed Cq values, and fold changes were calculated using the ΔΔCq method. hsa-miR-99a-5p and hsa-miR-361-5p were selected as the most stable pair of miRNAs across the samples (stability value = 0.05) by NormFinder software (https://moma.dk/normfinder-software, accessed on 1 December 2019 and therefore used as the housekeeping genes. 

#### 2.2.2. Statistical Analysis

Fold change values were tested for normality and found to not be normally distributed using the Shapiro–Wilk test. Mann–Whitney U tests was used to test differences in miRNA expression between two groups. To check for variability across different groups, a heat map analysis was performed.

#### 2.2.3. MiRNA Targets and Pathways Analysis

Diana tools mirPath v.3 (http://snf-515788.vm.okeanos.grnet.gr/, accessed on 20 June 2021) was used for the downstream analysis. The DIANA-microT-CDS algorithm with a set threshold of 0.8 was used to select mRNA targets on the gene union tool. The final *p*-value was corrected with FDR. A *p*-value of less than 0.05 was considered significant. KEGG (Kyoto Encyclopedia of Genes and Genomes, Kyoto, Japan) pathways analysis was also performed.

### 2.3. Protein Analysis

#### 2.3.1. Protein Extraction and Analysis

Proteins were extracted with scioExtract buffer (Sciomics, Neckargemünd, Germany) using the extraction SOPs. The bulk protein concentration was determined by BCA assay. The reference samples and the samples were labelled for two hours with scioDye 1 and scioDye 2, respectively. After that the reaction was stopped, the buffer exchanged to PBS and samples stored at −20 °C until use.

Samples were analysed in a dual-colour approach using a reference-based design on scioDiscover antibody microarrays (Sciomics, Neckargemünd, Germany) targeting 1351 different proteins with 1821 antibodies in four replicates. The arrays were blocked with scioBlock (Sciomics, Neckargemünd, Germany) on a Hybstation 4800 (Tecan, Männedorf, Switzerland), and afterwards the samples were incubated competitively with the reference sample using a dual-colour approach. After incubation for three hours, the slides were thoroughly washed with 1× PBSTT, rinsed with 0.1× PBS and water, and finally dried with nitrogen.

#### 2.3.2. Data Acquisition and Statistical Analysis

Slide scanning was conducted using a Powerscanner (Tecan, Männedorf, Switzerland) with constant instrument power and PMT settings. Spot segmentation was performed with GenePix Pro 6.0 (Molecular Devices, Union City, CA, USA). Data were analysed using the linear models for microarray data (LIMMA) package of R-Bioconductor. For the normalisation, a specialised invariant Lowess method was applied. For analysis of the samples, a one-factorial linear model was fitted with LIMMA resulting in a two-sided *t*-test or F-test based on moderated statistics. All presented *p*-values were adjusted for multiple testing by controlling the false discovery rate according to Benjamini and Hochberg. Proteins were defined as deferential for ׀logFCj׀ > 0.5 and an adjusted *p*-value < 0.05. Proteins differentially expressed between groups are presented as log-fold changes (logFC) calculated for the basis 2. When samples versus controls are compared, a logFC = 1 means that the sample group had on average a 2^1^ = 2-fold higher signal than the control group. logFC = −1 stands for 2^−1^ = 1/2 of the signal in the sample as compared to the control group.

#### 2.3.3. Pathway Analysis

Proteins were defined as significantly differential for |log2FC| ≥ 0.5 and a simultaneous adjusted *p*-value ≤ 0.05. All differential proteins were subjected to STRING (search tool for the retrieval of interacting genes/proteins) analysis for the visualization of protein networks, pathway analysis (based on KEGG and Reactome databases), as well as biological function classification analysis (based on the gene ontology (GO) database). KEGG pathways only are shown in this paper.

### 2.4. Pathway Studio

As part of our pathway analysis and gene ontology representation, we utilized the Elsevier’s PathwayStudio v. 10 (https://www.elsevier.com/solutions/pathway-studio-biological-research, accessed on 20 June 2021). This software handles high throughput “omics” data that can be interrogated for functional analysis and interactome assessment. For this purpose, differential proteins and altered miRNAs were evaluated using the propriety PathwayStudio ResNet database gene analysis. “Subnetwork Enrichment Analysis” (SNEA) algorithm builds functional pathways where Fisher’s statistical test is used to associate one differential hit (gene/protein/microRNA, etc.) to a specific molecular function, pathway, or biological process with the option of localizing these in cellular compartments. The association is categorized into protein modification targets, regulation targets binding partners, and expression targets.

## 3. Results

### 3.1. Demographic Characteristics

The demographic characteristics are reported in Table 1. For each group, 3F and 2M samples were selected. The average age ± standard deviation for the control group was 85.4 ± 9.2; 82 ± 4.5 and 84.4 ± 6.8 for Braak III-IV and Braak V-VI respectively. The average post-mortem delay ± standard deviation for control group was 49.7 ± 13.00; 35.8 ± 20.18 and 43.25 ± 17.68 for Braak III-IV and Braak V-VI respectively. History of diagnosis, Braak tangle stage, and CERAD (Consortium to establish a registry for Alzheimer’s disease) are also reported in Table 1.

### 3.2. MicroRNA Analysis

Fifty-four differentially expressed miRNAs (DE-miRNAs) were found, of which 14 were substantially different between control and Braak stage III-IV and 44 between control and Braak stage V-VI (Table 2). A further analysis comparing Braak stage III-IV versus Braak stage V-VI is available in Appendix A.

These changes can also be visualised using a hierarchal clustered heatmap (Figure 1). There is a clear shift towards blue in the Braak stage V-VI, whereas the control group is red, with the Braak stage III-IV sharing differences with both sides airing closer to the control group. Overall, this visual representation demonstrates that the majority of differentially expressed miRNAs are downregulated as the disease progresses.

### 3.3. Protein Analysis

Upon extraction, one sample (Braak stage III-IV) featured a slightly red colour indicating either a higher blood content or some impurities in the extraction process. In addition, this sample was tagged in the quality control analysis due to lower overall signal intensities. Therefore, this sample was excluded from the biostatistical analysis. 

Fold changes and statistically significant proteins are listed in Table 3. 

Comparing the AD subtypes with controls, 73 differential proteins were identified for Braak stage III-IV and 119 for Braak stage V-VI. In addition, 18 proteins were differential for both AD subtypes.

A further comparison (Braak stage III-IV vs. Braak stage V-VI) is available in Appendix A.

Relative expression levels for differential proteins identified across all comparisons with both ׀logFC׀ > 0.5 and adj. *p*-value < 0.001 are summarised in heat map Figure 2.

### 3.4. MiRNA Target and Pathways Analysis

MiRNA targets of DE-miRNA were selected using DIANA tools as described above. The predicted target genes for DE-miRNAs were compared against differentially expressed proteins (DE-proteins) found in the brain samples. Inverse relationships between expression of miRNA and target proteins were checked and showed one case in Braak stage III-IV and 34 cases in Braak stage V-VI (Table 4).

Pathways analyses obtained for both DE-miRNAs and DE-proteins were compared and resulted in one common pathway (Rap1 signalling pathway) in Braak stage III-IV (Figure 3) and six pathways (ECM-receptor interaction, MAPK signalling pathway, Ras signalling pathway, PI3K-Akt signalling pathway, FoxO signalling pathway, and Rap1 signalling pathway) in Braak stage V-VI (Figure 4).

Pathways analyses obtained for both DE-miRNAs and DE-proteins in Braak stage III-IV.showed one common pathway (Rap1 signalling pathway).

Pathways analyses obtained for both DE-miRNAs and DE-proteins in Braak stage V-VI. showed six common pathways (ECM-receptor interaction, MAPK signalling pathway, Ras signalling pathway, PI3K-Akt signalling pathway, FoxO signalling pathway, and Rap1 signalling pathway).

In Rap1 signalling pathway of Braak stage III-IV, no inverse relation was found between miRNA targets and proteins analysed in this study. In the six common pathways identified in Braak stage V-VI, the inverse relations between DE-miRNAs and DE-proteins are reported in Table 5.

Some of these inverse relationships were verified by using ELISA kits (all details are presented in Appendix A).

### 3.5. Pathway Studio

Pathway studio was interrogated to analyse DE-miRNAs and DE-proteins at early and late stage of AD. Figure 5 panel A shows the pathological processes triggered by Braak III-IV DE-miRNA. The most relevant AD pathways, namely lipotoxicity, apoptosis, and senescence, are shown in panel B of the same figure. The senescence pathway showed to be regulated by miR-204, one of the most significantly expressed miRs of early AD. In panel C and D, the pathological processes of miRNA targets and the main AD Braak V-VI pathway-related ones, including cell death, cell ageing, senescence, and apoptosis are shown, respectively.

DE-proteins pathways are represented in Figure 6 and confirmed the ECM and cell-cell adhesion among the main processes regulated during AD development.

## 4. Discussion

Alzheimer’s disease is a complex, multifactorial disease with no available treatment due to the lack of understanding of causative factors of the disease. It is also difficult to diagnose AD early in the disease process because of the late onset nature of symptoms and the lack of clinical clarity between AD and other neurodegenerative and neurocognitive diseases. Understanding the mechanisms underlying the initiation of neurodegeneration is critical, and the early stages of disease may present a valuable therapeutic window before irreversible brain damage has occurred. Therefore, in this paper, we had two main objectives; the first one was to identify the molecular mechanisms and the cellular processes developed in the late and confirmed cases of AD. We achieved this outcome profiling microRNAs and proteins in the brains of AD Braak V-VI patients. The microRNAs and proteins found significantly and differentially expressed were then used for advanced bioinformatic analyses to determine their relative pathways. In addition, an inverse relation between miRs and protein targets was also investigated. The inverse association with protein targets has a greater likelihood of directly targeting those proteins, making them suitable for miRNA therapy.

The second objective was to perform the same analyses as above using brain tissue of AD patients at an early stage (Braak III-IV) and comparing them to controls. This second outcome was to evaluate if any of the mechanisms detected in the late phase of AD could be earlier identified, therefore opening a window for potential early interventions.

MiRNAs were a phenomenon discovered in 1993, and ever since, there has been much interest in their ability to regulate gene expression within diseases [30]. More recently, they have been studied as biomarker candidates of many pathologies, including AD, in blood and CSF [31,32]. A comparison between blood/CSF miRNAs and brain DE-miRNAs of this research is available in Appendix A. Results matched 14 DE-miRNAs between biofluids and brain tissue, offering an alternative way of non-invasively monitoring any change or advancement of AD pathology. 

In this study, we identified 14 DE-miRNAs and 73 DE-proteins between the control and early stage of AD samples. An inverse expression was seen in the upregulation of miR-204-5p and its potential downregulated target SELE (selection E), a protein responsible for accumulation of leukocytes at the sites of inflammation. MiR-204 was also depicted by our analysis using pathway studio as key regulator of senescence (Figure 5, panel B). Other studies showed miR-204-5p to be one of the most abundantly expressed in AD [33] and upregulated in sporadic Parkinson’s disease patients, leading to apoptosis of dopaminergic cells in the brain [34]. However, none was able to prove that SELE was a miR-204-5p target.

Bioinformatic analysis revealed 58 mutual KEGG pathways between Braak III-IV DE-miRNAs and DE-proteins and one common pathway, the R1 signalling pathway. Rap1 is a small GTPase that controls diverse processes, such as cell adhesion, cell-cell junction formation, and cell polarity, by regulating the function of integrins and other adhesion molecules [35,36]. Previous studies showed that inhibition of Rap1 interactions reduces Ca^2+^ dysfunction and improves neuronal survival [37]. 

Cell adhesion and extracellular matrix (ECM) degradation are two pathways found regulated by DE-proteins of early AD using pathway studio. The same pathways were also found in DE-miRNAs and DE-proteins of the late stage of AD.

In the late stage of AD, we reported 44 DE-miRNAs compared to control, of which four were in common with the early stage of AD, and 119 DE-proteins, of which 18 were in common with the early stage. By identifying miRNAs and target proteins with an inverse expression, we were also able to select 34 potential connections that may play a role in AD pathology (Table 4). The associated KEGG pathways also helped to understand the possible cellular effects of each deregulated miRNA. Forty-two pathways were found in proteins and 69 in microRNAs and six pathways were in common: ECM-receptor interaction, MAPK signalling pathway, Ras signalling pathway, PI3K-Akt signalling pathway, FoxO signalling pathway, and Rap1 signalling pathway.

The first pathway, the ECM-receptor interaction, also confirmed by the pathway studio analysis of DE-proteins, is related to cellular integrity and consists of proteoglycans/glycosaminoglycans (PGs/GAGs), proteins, proteinases, and cytokines, which surround cells and facilitate intercellular communication [38]. In CNS, fibrous proteins (collagen, elastin) and adhesive glycoproteins (laminin, fibronectin) were responsible for the strength and elasticity of the ECM [39,40]. ECM also regulates intercellular communication and coordinate processes, such as cell migration, proliferation, differentiation, and apoptosis as well as tissue morphogenesis and homeostasis [41,42], and aberrant expression of the ECM components in the brain can lead to neuro-development disorders, psychiatric dysregulation, and neuro-degenerative diseases [43,44]. An upregulation of collagen IV, laminin, and fibronectin in the brains of AD and a co-localization with Senile Plaques was also proven [45].

In our study, we found several inverse expressions between miRs and proteins with a function in the ECM pathway. Among them, we found hsa-miR-671-5p and Thrombospondin 1 (THBS1), an adhesive glycoprotein that mediates cell-to-cell and cell-to-matrix interactions. This protein can bind to fibrinogen, fibronectin, laminin, type V collagen, and integrins alpha-V/beta-1, hsa-miR-132-3p, and hsa-miR-212-3p, inversely related to CD44, which was previously found upregulated in severe Alzheimer’s disease [46]. hsa-miR-30e-5p and Integrin beta-1 (ITGB1), a receptor for collagen, is another interaction, and it was suggested to undergo plasticity through interactions with ECM proteins, modulating ion channels, intracellular Ca^2+^ and protein kinases signalling, and reorganization of cytoskeletal filaments [47]. ECM and integrin aberrations are likely to contribute to imbalanced synaptic function in epilepsy, Alzheimer’s disease, mental deficiency, schizophrenia, and other brain conditions [48]. hsa-miR-1-3p, hsa-miR-425-3p, and hsa-miR-1237-3p are all downregulated, and all show the upregulated fibronectin (FN1) as potential target. Fibronectin is the adhesive protein that connects cells with collagen fibres in the ECM, binds to cell-surface integrin regulating cytoskeleton reorganization, and facilitates cell migration [48,49,50,51,52,53]. 

In addition, several studies reported that the upregulation of fibronectin enhanced the adherence of microglial cells and lead to APP secretion [54,55] and Aβ aggregation in AD pathogenesis. 

The other five pathways found to be compromised in the late stage of AD are MAPK, FoxO, Rap1, Ras, and PI3K-Akt signalling pathways. These are intracellular signalling pathways and are required for signal relay, signal amplification, and integration of stimuli in the cell, leading to proliferation, differentiation, apoptosis, and inflammation. 

The MAPK signalling pathway is key for signal transduction between neurons and is involved in the maintenance of synaptic plasticity, which is reduced with progression of AD [56]. In particular, p38 MAPK is stimulated by phosphorylated tau, normal dephosphorylated tau, and Aβ oligomers, leading to inflammatory signalling and activation of microglia in AD and inhibition of p38 MAPK pathways, successfully reducing Aβ and memory loss in AD mice [56,57,58,59,60] and suggesting this MAPK signalling as an important therapeutic target. 

FoxO signalling pathway is active in many parts of the brain, including the hippocampus, amygdala, and nucleus accumbens. It functions to determine cell fate and survival, leading to apoptosis, cell proliferation, and differentiation. Dysregulation of FoxOs can lead to increased apoptosis and oxidative stress [61,62]. Some studies also suggest that FoxO proteins may be a downstream target of APP, leading to an APP intracellular domain induced death in neurones, with APP as a transcriptional coactivator of FoxO proteins [63]. Other studies speculated that FoxO proteins may also be upstream regulators of APP [64].

The phosphoinositide 3-kinase (PI3K)/protein kinase B (Akt) pathway seems to be particularly important for mediating neuronal survival [65], playing a role in learning and memory as well [66,67]. It is also involved in the pathogenesis of acute cerebrovascular disease, neurodegeneration diseases, epilepsies, and AD [68,69]. The epidermal growth factor receptor (EGFR) was one of the inverse expressed targets in this pathway, and several studies demonstrated that EGFR inhibitors improve pathological and behavioural conditions in AD [70,71]. Moreover, EGFR expression is up-regulated in reactive astrocytes, and its hyper-activation leads to astrogliosis [72,73]. The neuroprotective effect of EGFR inhibition was demonstrated in ALS also [71]. EGFR is regulated by miR-1-3p, and this interaction was already found in cancer [74]. 

Finally, research groups also reported an increased Ras expression in cells with upregulated APP expression, which could lead to deregulation of the cell cycle [75].

### Limitations

All samples used in this study were collected within 72 h of death and then frozen. This long interval of time could have affected the miRNA abundance due to the degradation of the miRNAs. The sample size for all groups (*n* = 5 per group) was small, and the control and AD groups decreased to *n* = 4 for the protein microarray. Due to the exploratory nature and the small sample size of the study, differences between groups in miRNAs are presented without adjustments for multiple comparisons; thus, further studies with larger numbers of samples will be required to assess the reproducibility of these findings.

Moreover, we were unable to test for specific proteins in the microarray because of the Sciomics customised platform (1351 proteins only) for this analysis. As result, not all specific targets of the differential miRNAs have been analysed, and the blind analysis of proteins limited the comparison to miRNA expression in this study; therefore, it is important to look at specific protein targets in the future. 

Finally, of the 2300 known miRNAs in the human brain, only 751 were analysed for differential expression in this study. This will be an important avenue for future investigation.

## 5. Conclusions

In conclusion, our results demonstrate that miRNAs and proteins are differentially expressed in the early and late stages of Alzheimer’s disease. The correlation between their expression must be explored further to understand how these miRNAs regulate gene expression in disease progression. In addition, mutual signalling pathways between the differential miRNAs and proteins were investigated, showing only one common pathway (the Rap1 signalling) between the early and late stage of AD. 

While the differentially expressed miRNAs and mutual signalling pathways that were found in this study are in line with previous work, new interactions between miRNA and protein targets were also identified. These novel findings must be investigated further to gain a deeper understanding of their roles in AD and their potential use as therapeutic targets. The results of this can be taken as a basis for future, larger-scale studies. 

## Figures and Tables

**Figure 1 cells-11-00163-f001:**
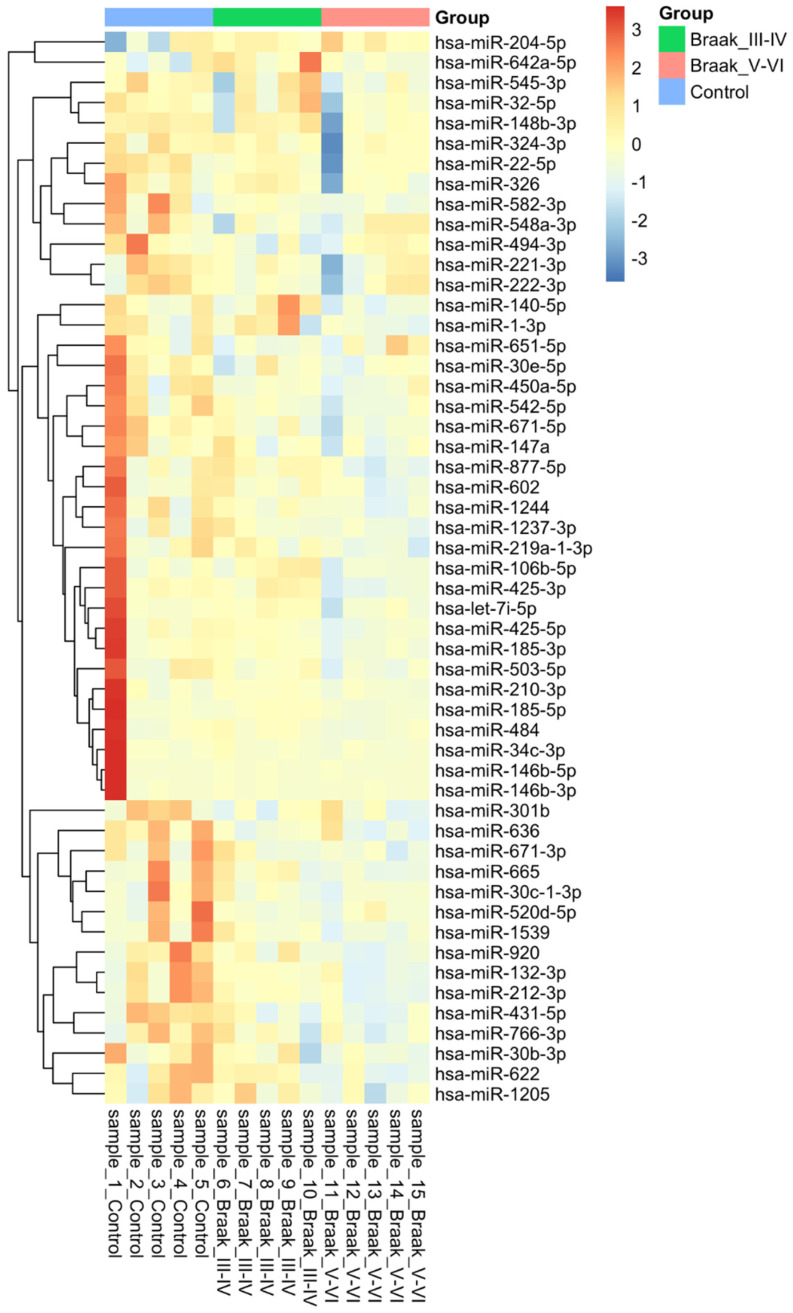
Heatmap showing the expression of 54 differentially regulated microRNAs in Alzheimer’s disease. Fifty-four microRNAs were found using Mann–Whitney statistical analysis across the 2 different comparisons, Control vs. Braak stage III-IV and Control vs. Braak stage V-VI. A *p*-value of less than 0.05 was considered significant. Hierarchal clustering is present on the left with colours relevant to their group. This was made using heatmap.2 in the R programming language with complete linkage, and Euclidean distance used to compute.

**Figure 2 cells-11-00163-f002:**
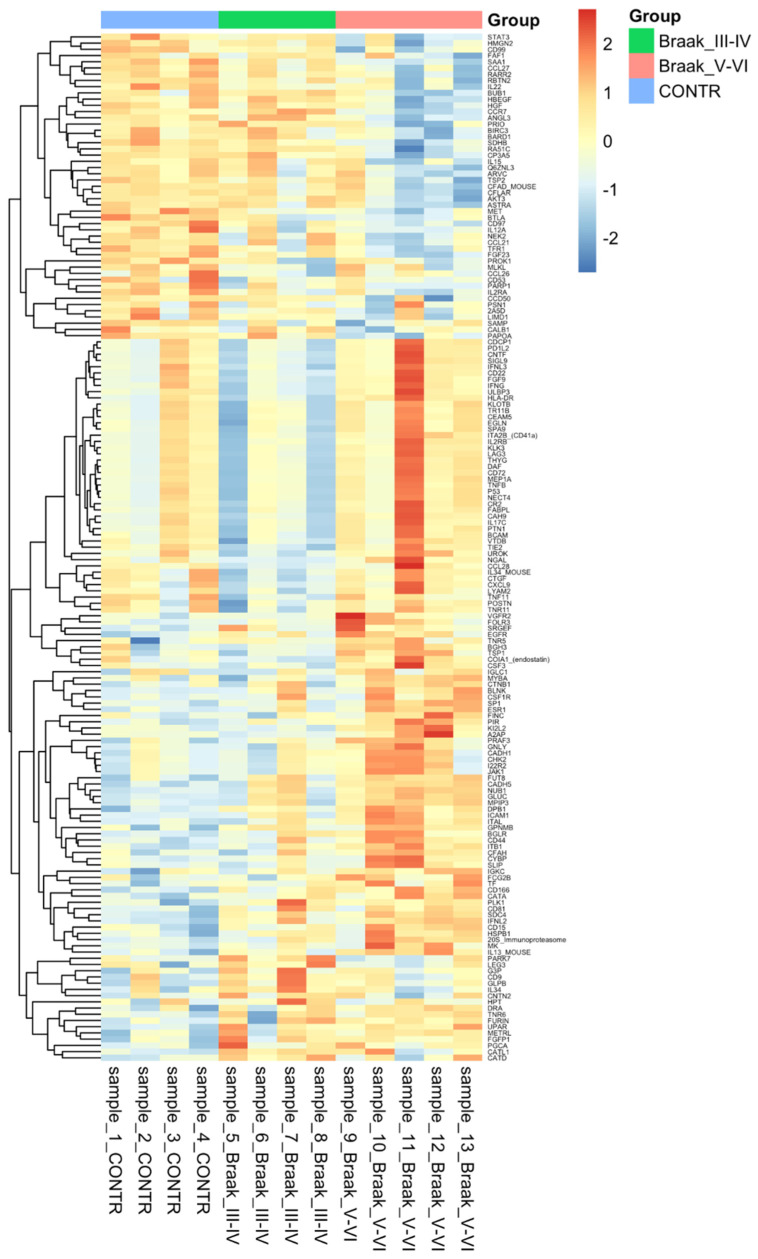
Heatmap showing the expression of 174 differentially regulated proteins in Alzheimer’s disease across the 2 compariosns Control vs. Braak stage IiI–IV and Control vs. Braak stage V-VI. A *p*-value of less than 0.05 was considered significant. Hierarchal clustering is present on the left with colours relevant to their group. This was made using heatmap.2 in the R programming language with complete linkage, and Euclidean distance used to compute.

**Figure 3 cells-11-00163-f003:**
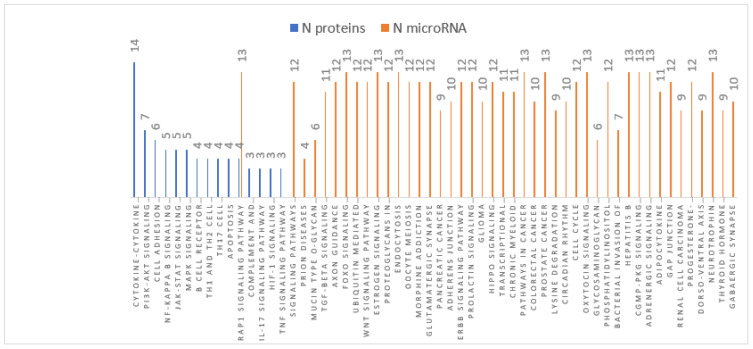
Braak stage III-IV pathways analyses.

**Figure 4 cells-11-00163-f004:**
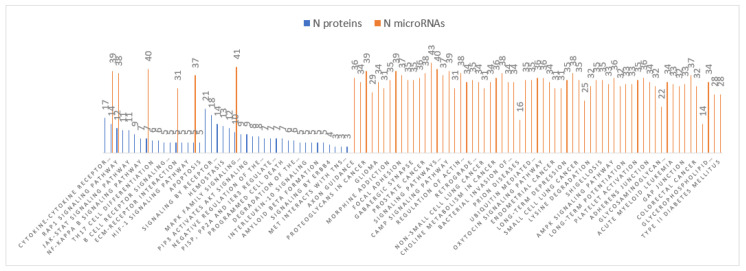
Braak stage V-VI pathways analyses.

**Figure 5 cells-11-00163-f005:**
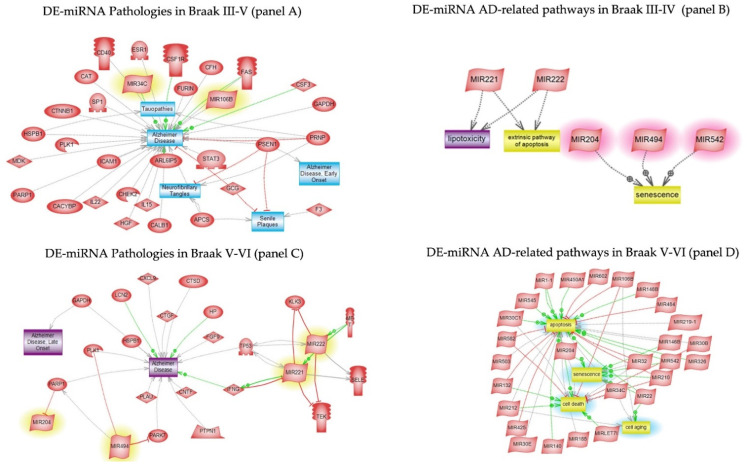
DE-miRNA pathway studio analyses in Braak III-IV and Braak V-VI. The pathological processes triggered by Braak III-IV DE-miRNA (panel **A**) and Braak V-VI DE-miRNA (panel **C**). The related pathways of Braak III-IV DE-miRNA (panel **B**) and Braak V-VI DE-miRNA (panel **D**).

**Figure 6 cells-11-00163-f006:**
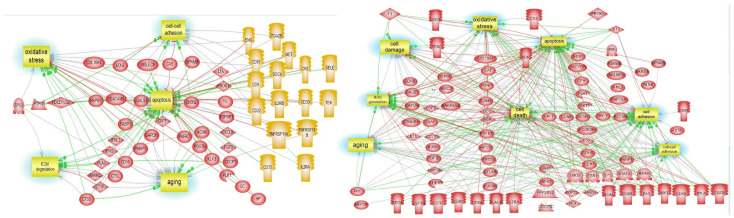
DE-protein pathway studio analyses in Braak III-IV and Braak V-VI. AD pathways generated by DE-proteins in Braak III-IV (panel **A**) and Braak V-VI DE-miRNA (panel **B**).

**Table 1 cells-11-00163-t001:** Demographics of samples received from South-West Dementia Brain Bank. CAA= cerebral amyloid angiopathy.

Groups	Age	Sex	Post-Mortem Delay	Hist Diagnosis	Braak Tangle Stage	CERAD Diag. Group
Controls	77	M	42	No AD, mild to moderate CAA	1	No AD
	74	F	39.5	Control, no significant abnormalities	1	No AD
	92	M	56.5	Control, moderate CAA	1	No AD
	90	F	67.25	Single microinfarct in occipital cortex fine to use as control	1	No AD
	94	F	43.25	Normal-looking brain	1	No AD
Early stage	79	F	70	AD	3	
	90	F	21	AD	4	
	80	M	24	AD probable, moderate CAA	4	Probable AD
	80	F	26	AD probable, moderate CAA	4	Probable AD
	81	M	38	AD definite	4	Definite AD
Late stage	88	F	68	AD definite, moderately severe CAA	5	Definite AD
	93	F	31.75	AD definite	5	Definite AD
	78	M	49.5	AD mod severe	6	Definite AD
	86	F	45.25	AD definite (severe)	6	Definite AD
	77	M	21.75	AD definite, moderate CAA	6	Definite AD

**Table 2 cells-11-00163-t002:** DE-miRNA in Braak stage III-IV and Braak stage V-VI compared to controls (Braak stage I-II). Fold changes (FD), standard deviation (SD), and *p*-values ≤ 0.05 are reported.

MiRNAs	CONTR FC ± SD	Braak III-IV FC ± SD	CONTR vs. Braak III-IV *p*-Value	Braak V-VI FC ± SD	CONTR vs. Braak V-VI *p*-Value
hsa-miR-221-3p	1.01 ± 0.15	0.76 ± 0.11	0.03		
hsa-miR-431-5p	1.01 ± 0.17	0.63 ± 0.23	0.03	0.55 ± 0.14	0.01
hsa-miR-494-3p	1.01 ± 0.20	0.65 ± 0.20	0.03		
hsa-miR-671-5p	1.02 ± 0.21	0.66 ± 0.17	0.03	0.52 ± 0.15	0.01
hsa-miR-222-3p	1.01 ± 0.15	0.76 ± 0.08	0.03		
hsa-miR-542-5p	1.06 ± 0.37	0.56 ± 0.13	0.03	0.39 ± 0.18	0.01
hsa-miR-642a-5p	1.10 ± 0.50	3.79 ± 2.87	0.03		
hsa-miR-548a-5p	1.03 ± 0.24	0.40 ± 0.27	0.01		
hsa-miR-204-5p	1.04 ± 0.30	1.50 ± 0.12	0.01	1.66 ± 0.37	0.01
hsa-miR-520d-5p	1.22 ± 0.83	0.45 ± 0.15	0.03		
has-miR-651-5p	1.05 ± 0.37	0.53 ± 0.17	0.03		
has-miR-301b-3p	1.02 ± 0.22	0.63 ± 0.20	0.03		
hsa-miR-636	1.04 ± 0.31	0.37 ± 0.13	0.01		
hsa-miR-548a-3p	1.05 ± 0.36	0.47 ± 0.32	0.03		
hsa-miR-140-5p	1.00 ± 0.07			0.81 ± 0.09	0.01
hsa-miR-210-3p	1.19 ± 0.93			0.54 ± 0.16	0.03
hsa-miR-185-5p	1.09 ± 0.59			0.64 ± 0.05	0.01
hsa-miR-425-5p	1.03 ± 0.33			0.61 ± 0.10	0.01
hsa-miR-503-5p	1.13 ± 0.64			0.37 ± 0.19	0.01
hsa-miR-30b-3p	1.02 ± 0.22			0.60 ± 0.12	0.01
hsa-miR-545-3p	1.00 ± 0.11			0.68 ± 0.19	0.03
hsa-miR-665	1.09 ± 0.53			0.40 ± 0.12	0.01
hsa-miR-132-3p	1.05 ± 0.33			0.47 ± 0.23	0.03
hsa-miR-22-5p	1.01 ± 0.13			0.64 ± 0.30	0.01
hsa-miR-30e-5p	1.00 ± 0.12			0.76 ± 0.09	0.01
hsa-miR-32-5p	1.00 ± 0.10			0.67 ± 0.29	0.01
hsa-miR-324-3p	1.00 ± 0.07			0.81 ± 0.23	0.03
hsa-miR-877-5p	1.01 ± 0.20			0.67 ± 0.13	0.03
hsa-let-7i-5p	1.04 ± 0.35			0.64 ± 0.20	0.01
hsa-miR-602	1.06 ± 0.43			0.57 ± 0.20	0.03
hsa-miR-148b-3p	1.00 ± 0.02			0.77 ± 0.22	0.01
hsa-miR-34c-3p	1.48 ± 1.76			0.39 ± 0.18	0.01
hsa-miR-622	1.12 ± 0.48			0.38 ± 0.27	0.01
hsa-miR-185-3p	1.10 ± 0.62			0.43 ± 0.15	0.01
hsa-miR-766-3p	1.00 ± 0.12			0.78 ± 0.12	0.03
hsa-miR-326	1.02 ± 0.21			0.63 ± 0.29	0.03
hsa-miR-147a	1.05 ± 0.37			0.45 ± 0.27	0.03
hsa-miR-146b-5p	1.84 ± 2.72			0.41 ± 0.09	0.01
hsa-miR-1-3p	1.01 ± 0.18			0.69 ± 0.11	0.03
hsa-miR-425-3p	1.01 ± 0.18			0.66 ± 0.07	0.01
hsa-miR-450a-5p	1.07 ± 0.38			0.48 ± 0.25	0.03
hsa-miR-106b-5p	1.02 ± 0.26			0.72 ± 0.11	0.01
hsa-miR-212-3p	1.06 ± 0.38			0.36 ± 0.18	0.01
hsa-miR-484	1.06 ± 0.47			0.66 ± 0.09	0.01
hsa-miR-219a-1-3p	1.04 ± 0.35			0.48 ± 0.15	0.01
hsa-miR-1244	1.05 ± 0.36			0.48 ± 0.14	0.01
hsa-miR-1205	1.04 ± 0.31			0.60 ± 0.29	0.03
hsa-miR-146b-3p	1.78 ± 2.56			0.42 ± 0.08	0.01
hsa-miR-920	1.09 ± 0.48			0.20 ± 0.11	0.01
hsa-miR-1237-3p	1.10 ± 0.52			0.47 ± 0.13	0.03
hsa-miR-671-3p	1.016 ± 0.20			0.67 ± 0.11	0.01
hsa-miR-582-3p	1.03 ± 0.29			0.61 ± 0.07	0.01
hsa-miR-1539	1.12 ± 0.61			0.36 ± 0.17	0.01
hsa-miR-30c-1-3p	1.02 ± 0.21			0.74 ± 0.09	0.01

**Table 3 cells-11-00163-t003:** DE-proteins in Braak stage III-IV and Braak stage V-VI compared to controls (Braak stage I-II). Fold changes (LogFC), standard deviation, and adj. *p*-values ≤ 0.05 are reported.

Protein	HGNC Number	Braak Stage III-IV vs. CONTR logFC	adj. *p*-Value	Braak Stage V-VI vs. CONTR logFC	adj. *p*-Value
G3P	GAPDH	1.4	0.0074	0.97	0.044
DPB1	HLA-DPB1	1.23	0.0061	2.05	1.9 × 10^−6^
METRL	METRNL	1.2	0.012	1.31	0.0038
CD9	CD9	1.15	0.0057		
IL34	IL34	1.07	0.02		
GLPB	GYPB	1.03	0.0061		
FGFP1	FGFBP1	0.92	0.043		
HPT	HP	0.8	0.048		
GPNMB	GPNMB	0.79	0.019	0.95	0.0025
CD81	CD81	0.74	0.0029	0.71	0.0017
PLK1	PLK1	0.65	0.033	0.61	0.042
CNTN2	CNTN2	0.62	0.022		
SDC4	SDC4	0.62	0.0044	0.85	0.000014
CATL1	CTSL	0.61	0.00025		
PARK7	PARK7	0.6	0.0017		
CATD	CTSD	0.55	0.00026		
PGCA	ACAN	0.55	0.021	0.55	0.016
LEG3	LGALS3	0.54	0.018		
HSPB1	HSPB1	0.51	0.033	0.77	0.00051
UROK	PLAU	−0.54	0.032		
CR2	CR2	−0.56	0.033	1.07	0.041
TR11B	TNFRSF11B	−0.59	0.032		
TNFB	LTA	−0.59	0.017		
FGF9	FGF9	−0.6	0.018		
MEP1A	MEP1A	−0.6	0.011		
CAH9	CA9	−0.61	0.0061		
EGLN	ENG	−0.61	0.02		
FABPL	FABP1	−0.61	0.011		
KLOTB	KLB	−0.61	0.019		
IL17C	IL17C	−0.62	0.021		
MLKL	MLKL	−0.63	0.0052		
CEAM5	CEACAM5	−0.63	0.018		
ITA2B (CD41a)	ITGA2B	−0.64	0.02		
IFNL3	IFNL3	−0.64	0.0079		
P53	TP53	−0.65	0.015		
PTN1	PTPN1	−0.66	0.0074		
SPA9	SERPINA9	−0.67	0.011		
NECT4	NECTIN4	−0.68	0.011		
IL2RB	IL2RB	−0.68	0.036		
CD22	CD22	−0.68	0.019		
DAF	CD55	−0.69	0.0061		
ULBP3	ULBP3	−0.69	0.0061		
NGAL	LCN2	−0.69	0.0059		
THYG	TG	−0.69	0.0095		
CD72	CD72	−0.7	0.0085		
VTDB	GC	−0.72	0.0044		
HLA-DR		−0.72	0.0059	1.53	0.0058
IFNG	IFNG	−0.72	0.036		
KLK3	KLK3	−0.73	0.0061		
EGLN	ENG	−0.74	0.0029		
TIE2	TEK	−0.74	0.013		
CDCP1	CDCP1	−0.75	0.005		
LAG3	LAG3	−0.76	0.0066		
BCAM	BCAM	−0.79	0.00035		
MET	MET	−0.82	0.019	−1.07	0.001
CNTF	CNTF	−0.83	0.0014		
PD1L2	PDCD1LG2	−0.84	0.0032		
IFNG	IFNG	−0.87	0.0061		
SIGL9	SIGLEC9	−0.91	0.011		
COIA1 (endostatin)	COL18A1	−0.94	0.0061	0.69	0.041
CXCL9	CXCL9	−0.98	0.048		
PROK1	PROK1	−1.01	0.0074	−0.76	0.042
PARP1	PARP1	−1.01	0.025	−1.19	0.0052
LYAM2	SELE	−1.05	0.012		
IL2RA	IL2RA	−1.11	0.011	−1.2	0.0037
CCL26	CCL26	−1.12	0.044		
CD53	CD53	−1.29	0.0081	−0.98	0.043
POSTN	POSTN	−1.33	0.011		
TNF11	TNFSF11	−1.4	0.0047		
IL34_MOUSE	Il34	−1.53	0.00014		
CCL28	CCL28	−1.57	0.000032		
CTGF	CTGF	−1.83	0.000055		
TNR11	TNFRSF11A	−1.9	0.00063		
CADH5	CDH5			3.25	0.00051
GLUC	GCG			2.36	4.8 × 10^−6^
NUB1	NUB1			1.98	0.00083
SP1	SP1			1.89	0.000024
CD44	CD44			1.62	0.000037
ICAM1	ICAM1			1.58	0.000013
I22R2	IL22RA2			1.56	0.011
ICAM1	ICAM1			1.49	0.000081
BLNK	BLNK			1.45	2.7 × 10^−6^
FUT8	FUT8			1.43	0.042
MK	MDK			1.43	0.00065
JAK1	JAK1			1.37	0.0055
ESR1	ESR1			1.37	1.9 × 10^−6^
IL13_MOUSE	Il13			1.32	0.0005
MPIP3	CDC25C			1.28	0.022
IGKC	IGKC			1.28	0.000017
CHK2	CHEK2			1.2	0.046
CSF1R	CSF1R			1.17	0.0023
KI2L2	KIR2DL2			1.13	0.0038
CYBP	CACYBP			1.09	0.0017
GNLY	GNLY			1.07	0.0037
CADH1	CDH1			1.07	0.035
BGH3	TGFBI			1.05	0.0032
VGFR2	KDR			1.04	0.015
PRAF3	ARL6IP5			0.99	0.011
UPAR	PLAUR			0.92	0.0094
MYBA	MYBL1			0.92	0.012
A2AP	SERPINF2			0.9	0.036
SRGEF	SERGEF			0.83	0.021
ITB1	ITGB1			0.83	5.4 × 10^−7^
CTNB1	CTNNB1			0.82	0.047
FOLR3	FOLR3			0.79	0.022
CSF3	CSF3			0.78	0.036
FCG2B	FCGR2B			0.76	0.00019
TSP1	THBS1			0.75	0.013
CD15				0.74	0.0012
TNR6	FAS			0.73	0.013
PIR	PIR			0.71	0.038
ITAL	ITGAL			0.69	0.00027
SLIP	NUGGC			0.69	0.025
TF	F3			0.66	1.1 × 10^−6^
TF	F3			0.66	1.9 × 10^−6^
EGFR	EGFR			0.66	0.0024
20S Immunoproteasome	PSMB8			0.65	0.005
CD166	ALCAM			0.65	0.0055
FINC	FN1			0.64	0.014
FURIN	FURIN			0.58	0.045
IFNL2	IFNL2			0.57	0.00018
CFAH	CFH			0.56	0.0094
BGLR	GUSB			0.55	0.0022
IGLC1	IGLC1			0.54	0.015
CD81	CD81			0.53	0.0023
CATA	CAT			0.52	0.0019
TNR5	CD40			0.52	0.005
TFR1	TFRC			−0.51	0.002
RBTN2	LMO2			−0.52	0.015
TFR1	TFRC			−0.53	0.00057
IL15	IL15			−0.54	0.043
CCL27	CCL27			−0.55	0.01
IL15	IL15			−0.56	0.049
CCD50	CCDC50			−0.56	0.000014
CALB1	CALB1			−0.58	0.049
BTLA	BTLA			−0.58	0.034
BUB1	BUB1			−0.59	0.014
HMGN2	HMGN2			−0.59	0.047
2A5D	PPP2R5D			−0.6	0.011
IL22	IL22			−0.61	0.03
STAT3	STAT3			−0.65	0.031
FAF1	FAF1			−0.65	0.039
SAMP	APCS			−0.66	0.03
CCR7	CCR7			−0.72	0.022
PRIO	PRNP			−0.74	0.0094
CD97	CD97			−0.79	0.001
HBEGF	HBEGF			−0.8	0.003
NEK2	NEK2			−0.82	0.0056
TFR1	TFRC			−0.82	0.0052
ANGL3	ANGPTL3			−0.83	0.013
AKT3	AKT3			−0.85	0.0089
PAPOA	PAPOLA			−0.89	0.046
IL12A	IL12A			−0.91	0.0071
RA51C	RAD51C			−1	0.0017
ASTRA	GRAMD1A			−1	0.0063
BIRC3	BIRC3			−1.01	0.02
RARR2	RARRES2			−1.01	0.00042
PSN1	PSEN1			−1.09	0.028
CD99	CD99			−1.09	0.026
BARD1	BARD1			−1.1	0.031
LIMD1	LIMD1			−1.17	0.00019
CP3A5	CYP3A5			−1.23	0.036
SDHB	SDHB			−1.24	0.017
CCL21	CCL21			−1.3	0.013
HGF	HGF			−1.31	0.017
PRIO	PRNP			−1.39	0.0018
SAA1	SAA1			−1.4	0.000006
TSP2	THBS2			−1.46	0.016
Q6ZNL3	FLJ00280			−1.56	0.038
CFAD_MOUSE	Cfd			−1.58	0.02
FGF23	FGF23			−1.7	1.2 × 10^−6^
ARVC	ARVCF			−1.85	0.012
CCR7	CCR7			−1.94	0.032
CFLAR	CFLAR			−2.27	0.001

**Table 4 cells-11-00163-t004:** Number of predicted targets for DE-miRNA in Braak III-IV and Braak V-VI and matched DE-proteins with an inverse relation.

MiRNAs	Number of Targets	Matched Protein Targets with Inverse Relation in Braak III-IV	Matched Protein Targets with Inverse Relation in Braak V-VI
hsa-miR-221-3p	302		
hsa-miR-431-5p	341		
hsa-miR-494-3p	1002		
hsa-miR-671-5p	414		THBS1, SP1
hsa-miR-222-3p	221		
hsa-miR-542-5p	16		
hsa-miR-642a-5p	53		
hsa-miR-548a-3p	26		
hsa-miR-204-5p	228	SELE	
hsa-miR-520d-5p	1538		
hsa-miR-651-5p	103		
hsa-miR-301b-3p	548		
hsa-miR-636	148		
hsa-miR-548a-3p	1504		
hsa-miR-140-5p	300		GPNMB
hsa-miR-210-3p	8		
hsa-miR-185-5p	802		ACAN, TGFB1
hsa-miR-425-5p	316		
hsa-miR-503-5p	276		
hsa-miR-30b-3p	553		ACAN, CDH1
hsa-miR-545-3p	235		
hsa-miR-665	391		HDR
hsa-miR-132-3p	626		CD44
hsa-miR-22-5p	575		
hsa-miR-30e-5p	1207		ITGB1, CAT
hsa-miR-32-5p	843		
hsa-miR-324-3p	91		
hsa-miR-877-5p	151		
hsa-let-7i-5p	646		IL13, FAS
hsa-miR-602	23		METRNL
hsa-miR-148b-3p	554		ALCAM
hsa-miR-34c-3p	276		SERPINF2, FURIN
hsa-miR-622	263		HLA-DPB1
hsa-miR-185-3p	263		HLA-DPB1
hsa-miR-766-3p	607		CHEK2
hsa-miR-326	349		ACAN
hsa-miR-147a	404		NUGGC
hsa-miR-146b-5p	682		MYBL1
hsa-miR-1-3p	634		EGFR, FN1
hsa-miR-425-3p	11		FN1
hsa-miR-450a-5p	24		
hsa-miR-106b-5p	1254		ESR1, FURIN, JAK1
hsa-miR-212-3p	597		CD44
hsa-miR-484	516		
hsa-miR-219a-1-3p	142		
hsa-miR-1244	233		
hsa-miR-1205	1011		CSF3
hsa-miR-146b-3p	270H		
hsa-miR-920	823		BLNK
hsa-miR-1237-3p	170		FNI
hsa-miR-671-3p	29		
hsa-miR-582-3p	340		
hsa-miR-1539	6		
hsa-miR-30c-1-3p	431		ACAN

**Table 5 cells-11-00163-t005:** Common pathways identified in DE-miRNAs and DE-proteins in Braak stage V-VI. For each of the pathways, the involved microRNAs and the correspondent predicted targets are listed. The protein targets identified in this study with an inverse relation are in bold.

Common KEGG Pathways in Braak V-VI	MicroRNAs	Targets in the Pathways and Inverse Relation
ECM-receptor interaction	hsa-miR-30b-3p	COL27A1, DAG1, COL11A2, VWF
	hsa-miR-671-5p	**THBS1**, CD47
	hsa-let-7i-5p	COL27A1, COL3A1, COL1A1, COL1A2, COL1A2, ITGA7, COL4A6, COL5A2, COL4A1
	hsa-miR-140-5p	DAG1, TNN
	hsa-miR-766-3p	LAMB4, COL24A1, DAG1
	hsa-miR-484	LAMB3, DAG1
	hsa-miR-920	ITGA3, DAG1
	hsa-miR-665	COL24A1, COL6A6, COL3A1, TNR
	hsa-miR-326	COL24A1, ITGA5, CD47
	hsa-miR-132-3p	ITGA9, COL4A4, COL11A1, COL5A2, **CD44**
	hsa-miR-30e-5p	**ITGB1**, ITGA9, ITGA8, ITGA4, ITGA6, ITGB3
	hsa-miR-148b-3p	LAMB2, ITGB8, ITGA9, ITGA5, ITGA11, COL2A1, COL4A1, LAMA4
	hsa-miR-212-3p	ITGA9, COL4A4, COL11A1, COL5A2, **CD44**
	hsa-miR-431-5p	ITGA1, LAMC2
	hsa-miR-1-3p	ITGB6, COL4A3, **FN1**, LAMC2
	hsa-miR-185-5p	SV2B, ITGA1
	hsa-miR-1237-3p	SV2B, ITGB6, COL6A6, THBS3, COL6A3, VWF, **FN1**, TNR
	hsa-miR-324-3p	ITGB5, COL2A1
	hsa-miR-622	ITGB5, COL2A1
	hsa-miR-106b-5p	ITGB8
	hsa-miR-1244	ITGB8, ITGAV
	hsa-miR-324-3p	ITGB5, COL2A1
	hsa-miR-622	ITGB5, COL2A1
	hsa-miR-106b-5p	ITGB8
	hsa-miR-1244	ITGB8, ITGAV
	hsa-miR-32-5p	ITGA8, COL27A1, ITGB6, ITGA5, ITGAV, COL5A1, COL1A2, COL11A1, ITGA6
	hsa-miR-204-5p	COL6A6, COL5A3
	hsa-miR-425-5p	TNR
	hsa-miR-147a	SV2A, ITGA11, TNR
	hsa-miR-582-3p	CD47
	hsa-miR-22-5p	ITGA5
	hsa-miR-425-3p	**FN1**
	hsa-miR-503-5p	COL4A1
MAPK signalling pathway	hsa-miR-503-5p	CACNA1I, RAPGEF2, FGF2, AKT3, FGF7
	hsa-miR-665	TGFBR1, MAPK8IP2, CACNG7, NLK, FGF2, MAX, AKT3, CACNA1D
	hsa-miR-132-3p	MAP3K3, CRK, KRAS, NLK, RASA1, ZAK, SOS1, TGFB2, AKT3, MECOM, DUSP9, MAPK1
	hsa-miR-212-3p	MAP3K3, CRK, KRAS, NLK, RASA1, ZAK, SOS1, TGFB2, AKT3, MECOM, DUSP9, MAPK1
	hsa-let-7i-5p	TGFBR1, MAP4K2, NRAS, ELK4, MAP4K3, MAP4K4, MAP3K1, TP53, FGF11, CASP3, RASGRP1, MAPK8, FLNA, NGF, **FAS**, MEF2C, DUSP1
	hsa-miR-34c-3p	MRAS, PAK2, BDNF, MAP2K6, PPP3CB, RASGRP1, PPM1A, SOS1, MAP3K2, MEF2C, GRB2
	hsa-miR-22-5p	PRKCA, ELK4, RAP1A, RAF1, MAP4K3, RASA1, SOS1, CACNA2D4, NF1, MAP2K1, MAP3K2
	hsa-miR-147a	CACNA1A, FLNC, FGF14, CRKL, MAP4K3, TAOK1, CACNA1I, FGF11, NLK, PDGFB, CACNA2D2, HSPA8, CACNG2, PRKCB, MAP3K2, DUSP9, FGFR2, MAPK10
	hsa-miR-1-3p	ATF2, RAP1A, MAP4K3, **EGFR**, MAP3K1, BDNF, IKBKB, RASA1, RAPGEF2, SOS1, MAX, NFATC3, RAP1B, PRKACB, PDGFA
	hsa-miR-1205	NTRK2, CACNA1A, MAP4K2, DUSP6, PLA2G4F, RASGRP2, IKBKB, JMJD7-PLA2G4B, CACNA1E, MAPT, NF1, MKNK1, RASGRP3, MEF2C, MAPK1, PDGFRB
	hsa-miR-30e-5p	RASA2, PPP3R1, MAP4K4, FGF20, MAP3K13, KRAS, TAOK1, PPP3CA, RASA1, CASP3, PPP3CB, RAPGEF2, MAPK8, SOS1, MAP3K2, RASGRP3, CACNB2, NFATC3, MAP3K7, RAP1B, MAP3K5, PDGFA, RPS6KA2
	hsa-miR-30b-3p	TAB1, MAPK8IP2, MAP3K13, PAK1, FGF11, DUSP10, PPP5D1, ZAK, ARRB2, CACNB2, MAPK1, ELK1
	hsa-miR-326	CACNG8, ELK4, RAF1, RPS6KA1, MRAS, PAK1, IKBKB, FGF9, FGF18, PTPN5, FGFR2, FGF1, ELK1
	hsa-miR-920	LAMTOR3, MAP3K2
	hsa-miR-431-5p	NTRK2, MAP3K13, TAOK1, RAPGEF2, CACNA1E
	hsa-miR-671-5p	NTRK2, NFKB1, STK4, MAP3K4, MAPKAPK3, MAPK1
	hsa-miR-106b-5p	TAOK3, BRAF, NTRK2, PDGFRA, DUSP2, RASA2, ELK4, CRK, MAPK7, RASGRF2, PPP3R1, MAP3K1, RRAS2, TAOK1, DUSP10, RASA1, MAPK8, ZAK, SOS1, STK3, HSPA8, NF1, MAP3K2, MKNK2, MAPK1, TGFBR2, MAP3K5, RPS6KA2
	hsa-miR-30c-1-3p	NTRK2, STK4, MAP3K13, PAK1, MAP2K6, FGF11, SOS1, ARRB1
	hsa-miR-545-3p	CRK, MAP3K1, TAOK1, PPP3CA, CACNA1B, PDGFB, FLNA, MAXPTPN5, MEF2C
	hsa-miR-148b-3p	SOS2, NRAS, GADD45A, MRAS, MAP3K4, MAP3K12, SOS1, MAX, TGFB2, DUSP1
	hsa-miR-146b-5p	BRAF, PDGFRA, HSPA1A, NRAS, TRAF6
	hsa-miR-766-3p	CACNA1G, RASGRP2, MAP3K4, BDNF, IKBKB, NLK, PRKCG, DUSP7, TNFRSF1A, RPS6KA4
	hsa-miR-425-5p	MAPK8, ZAK, SOS1, FGF9, PRKCB, STMN1, MEF2C, MAP3K5
	hsa-miR-219a-1-3p	FASLG
	hsa-miR-1244	CACNB4, RAPGEF2, MAPK8, MAP3K2, RAP1B
	hsa-miR-582-3p	CACNB4, MAP3K4, TAOK1, RAPGEF2
	hsa-miR-622	ATF2, RASA1, MAP3K12, MAPKAPK3, NF1, MECOM, STMN1
	hsa-miR-185-3p	PRKCA, CACNA1G, FLNC, MAP3K3, FGF17, MAPK13, RPS6KA5, PAK1, CACNA1I, FLNB, FLNA, MAPK1, CACNB3
	hsa-miR-484	RPS6KA1, RASGRP2, MAP3K11, FLNB, FLNA, FGF8, FGF1, MAPKAPK2, PDGFA
	hsa-miR-671-3p	MAP4K1, RASGRF1
	hsa-miR-185-5p	PAK2, CACNG7, MAPK14, MAP2K6, CACNG4, CDC42, PRKCB, RELA, NFATC3
	hsa-miR-146b-3p	RAF1, PRKCB, RPS6KA3, IL1A, PLA2G4C
	hsa-miR-204-5p	FGF20, TAOK1, SOS1, DUSP3
	hsa-miR-1237-3p	TAOK3, CACNG8, TGFBR1, PTPRR, TAB2, MAPK11, CACNA1E, STK3, FGF1, CACNB3
	hsa-miR-324-3p	TAB1, ARRB1
	hsa-miR-542-5p	IKBKB
	hsa-miR-140-5p	PDGFRA, GNG12, NLK, CACNB1, FGF9
	hsa-miR-32-5p	BRAF, RAP1A, CACNA1I, NLK, DUSP10, ZAK, MAP3K2, CACNA2D1, DUSP5, MAP2K4, RAP1B, DUSP1
	hsa-miR-877-5p	TAB2, MAP2K4
	hsa-miR-602	TAOK3, RASGRP2
	hsa-miR-450a-5p	CACNA1F, DUSP10
Ras signalling pathway	hsa-miR-1-3p	MET, RAP1A, ETS1, **EGFR**, GNB1, CALM2, IKBKB, RAB5A, RASA1, SOS1, IGF1, RAP1B, PRKACB, PDGFA
	hsa-miR-204-5	KSR2, SHC1, FGF20, SOS1
	hsa-miR-425-5p	MAPK8, SOS1, FGF9, PRKCB, GF1, ABL2, GRIN2B
	hsa-miR-132-3p	GNB1, KRAS, RASA1, SOS1, EXOC2, AKT3, PIK3CA, MAPK1, GRIN2B
	hsa-miR-22-5p	PRKCA, GNG11, PIK3CB, RAP1A, RAF1, RASA1, SOS1, NF1, MAP2K1, GRIN2B
	hsa-miR-30e-5p	RASA2, EFNA3, IGF1R, FGF20, KRAS, GNG10, RASA1, PIK3CD, PLA2G12A, MAPK8, BRAP, FLT1, SOS1, PDGFC, RASAL2, RASGRP3, RGL1, ABL2, GRIN2A, RAP1B, ABL1, PLA2G2C, PDGFA
	hsa-miR-148b-3p	RASAL1, SOS2, NRAS, ETS1, MRAS, PIK3R3, FLT1, SOS1, KITLG, IGF1, CSF1, TEK, GRIN2B
	hsa-miR-34c-3p	PAK2, MRAS, RASGRP1, REL, SOS1, GAB1, GRB2
	hsa-miR-106b-5p	PDGFRA, RASA2, RASGRF2, PAK7, TIAM1, PAK3, RRAS2, RASA1, MAPK8, REL, FLT1, SOS1, NF1, GAB1, SHC4, PDGFD, RGL1, MAPK1, ABL2, RAB5B, RAPGEF5, GNB5, GRIN2B
	hsa-miR-212-3p	GNB1, KRAS, RASA1, SOS1, EXOC2, AKT3, PIK3CA, MAPK1, GRIN2B
	hsa-miR-30c-1-3p	KSR2, STK4, PAK1, FGF11, SOS1, GRIN2A, RAB5B, GRIN2B
	hsa-miR-665	KSR2, RALA, FGF2, AKT3, **KDR**, GRIN2B
	hsa-miR-185-5p	KSR2, KSR1, PAK2, PAK7, GNB3, CDC42, PAK6, PRKCB, PAK6, RELA, VEGFA, ABL2, GRIN2B
	hsa-miR-1205	KSR2, MET, PIK3R2, PIK3R5, PLA2G4F, RASGRP2, IKBKB, JMJD7-PLA2G4B, PTPN11, NF1, PAK6, RASGRP3, RASSF1, MAPK1, PLA2G2C, PDGFRB, GRIN2B
	hsa-miR-484	KSR2, LAT, CALM1, RASGRP2, PAK3, PIK3CD, PDGFD, FGF8, GRIN1, FGF1, CSF1, PDGFA
	hsa-miR-185-3p	PRKCA, GNG13, FGF17, PLA2G6, PAK1, PAK6, SYNGAP1, MAPK1
	hsa-miR-30b-3p	KSR2, EFNA5, PAK1, PLA2G2D, FGF11, PAK4, PLD2, SYNGAP1, GAB2, RALB, MAPK1, RAB5B, ELK1, GRIN2B
	hsa-miR-766-3p	PIK3R5, ETS1, RASGRP2, IKBKB, PRKCG, PLD2, GRIN2B
	hsa-miR-326	GNGT1, RAF1, MRAS, PAK1, IKBKB, FGF9, PAK4, FGF18, RASAL2, FGFR2, FGF1, CSF1, ELK1, GRIN2B
	hsa-miR-147a	FGF14, CALM1, TIAM1, FGF11, PLCG1, PDGFB, PRKCB, PDGFC, FGFR2, MAPK10, GRIN2B
	hsa-miR-146b-5p	PDGFRA, NRAS, REL, ABL2, GRIN2B
	hsa-miR-1237-3p	KSR2, PAK7, GNG10, PLA2G12A, PIK3CA, RASAL2, HTR7, FGF1, GRIN2B
	hsa-miR-503-5p	IGF1R, PIK3R1, INSR, FGF2, AKT3, VEGFA, FGF7
	hsa-miR-324-3p	KSR2
	hsa-miR-146b-3p	KSR2, PAK7, RAF1, PAK3, MLLT4, PRKCB, PLA2G4C
	hsa-miR-920	KSR2, GAB1, SYNGAP1, CSF1
	hsa-miR-1244	RHOA, MAPK8, GAB1, HGF, RAP1B
	hsa-let-7i-5p	NRAS, IGF1R, FGF11, PLA2G3, RASGRP1, MAPK8, INSR, NGF, IGF1, PAK6, ABL2
	hsa-miR-542-5p	IKBKB
	hsa-miR-32-5p	GNG11, PIK3CB, RAP1A, PIK3R3, PIK3CA, RGL1, RAP1B
	hsa-miR-219a-1-3p	FASLG, IGF1
	hsa-miR-671-5p	NFKB1, STK4, SHC3, KIT, SYNGAP1, RAB5B, MAPK10
	hsa-miR-671-3p	RASGRF1
	hsa-miR-582-3p	PLD1, RHOA, PTPN11, ANGPT1
	hsa-miR-545-3p	PLA2G12A, PDGFB, PDGFD
	hsa-miR-622	RASA1, NF1
	hsa-miR-431-5p	RAB5B, RAPGEF5
	hsa-miR-140-5p	PDGFRA, GNG12, RALA, EFNA4, FGF9, BCL2L1
	hsa-miR-877-5p	PLD2
	hsa-miR-602	RASGRP2
PI3K-Akt signalling pathway	hsa-miR-30e-5p	**ITGB1**, ITGA9, ITGA8, EFNA3, PPP2R2B, IGF1R, FGF20, KRAS, IFNAR2, GNG10, DDIT4, PIK3CD, EIF4E, CCNE2, FLT1, SOS1, IL2RA, YWHAZ, IRS1, PDGFC, FOXO3, ITGA4, ITGA6, PPP2R1B, BCL2L11, PDGFA, ITGB3
	hsa-miR-1-3p	PRLR, MET, ATF2, PPP2R3A, CREB5, ITGB6, **EGFR**, GNB1, CDK6, YWHAQ, PPP2R5A, IKBKB, EIF4E, COL4A3, SOS1, YWHAZ, IGF1, **FN1**, LAMC2, PDGFA
	hsa-miR-582-3p	GSK3B, PPP2CA, CREB5, CDKN1B, YWHAQ, PTK2, YWHAZ, ANGPT1, FOXO3, PTEN
	hsa-miR-34c-3p	GSK3B, YWHAG, EIF4E, SOS1, GRB2
	hsa-miR-22-5p	PRKCA, MYB, GNG11, PIK3CB, ITGA5, RAF1, IL7R, PTK2, THEM4, SOS1, HSP90B1, PPP2R3C, PKN2, MAP2K1, SGK3
	hsa-miR-665	COL24A1, COL6A6, CDK6, COL3A1, FGF2, CASP9, AKT3, CDKN1A, TNR, PPP2R1B, **KDR**
	hsa-miR-766-3p	YWHAH, LAMB4, CDC37, COL24A1, PIK3R5, PPP2R5D, IKBKB
	hsa-miR-326	GNGT1, SYK, COL24A1, ITGA5, RAF1, PPP2R5D, IFNAR2, IKBKB, BRCA1, PPP2R5B, FGF9, FGF18, SGK2, GYS1, FGFR2, FGF1, CSF1
	hsa-miR-1244	ITGB8, LPAR3, ITGAV, HGF
	hsa-miR-146b-3p	RAF1, RHEB, RPS6KB2, LPAR5, CCNE1
	hsa-miR-32-5p	PHLPP2, TSC1, GNG11, PRKAA2, ITGA8, PIK3CB, COL27A1, ITGB6, ITGA5, PIK3AP1, GHR, ITGAV, DDIT4, PIK3R3, COL5A1, COL1A2, COL11A1, PIK3CA, ITGA6, PTEN, SGK3, BCL2L11
	hsa-miR-148b-3p	LAMB2, SOS2, ITGB8, ITGA9, NRAS, ITGA5, CDKN1B, YWHAB, ITGA11, COL2A1, PIK3R3, FLT1, SOS1, KITLG, PRKAA1, IGF1, PTEN, CSF1, TEK, BCL2L11, COL4A1, LAMA4
	hsa-let-7i-5p	PRLR, TSC1, MYB, NRAS, CCND2, COL27A1, IGF1R, COL3A1, RPS6KB2, TP53, GHR, FGF11, COL1A1, INSR, NGF, COL1A2, IGF1, ITGA7, COL4A6, CDKN1A, OSMR, COL5A2, COL4A1, IL6R
	hsa-miR-671-3p	RPS6KB2
	hsa-miR-1205	MET, PPP2CA, SYK, PIK3R2, PIK3R5, CCND2, MTCP1, IKBKB, TSC2, EIF4E, RPTOR, MAPK1, **CSF3**, BCL2L11, PDGFRB
	hsa-miR-132-3p	MYB, ITGA9, PRKAA2, CREB5, GNB1, PPP2R5C, KRAS, GHR, EIF4E, COL4A4, SOS1, AKT3, COL11A1, PIK3CA, FOXO3, PTEN, SGK3, MAPK1, COL5A2, C8orf44-SGK3
	hsa-miR-212-3p	MYB, ITGA9, PRKAA2, CREB5, GNB1, PPP2R5C, KRAS, GHR, EIF4E, COL4A4, SOS1, AKT3, COL11A1, PIK3CA, FOXO3, PTEN, SGK3, MAPK1, COL5A2, C8orf44-SGK3
	hsa-miR-1237-3p	PRKAA2, CDC37, CCND2, HSP90AA1, ITGB6, PPP2R5D, COL6A6, GNG10, IL4R, EIF4E, IL2RA, CHRM2, THBS3, COL6A3, PIK3CA, VWF, **FN1**, TNR, FGF1
	hsa-miR-484	CREB3L3, LAMB3, PPP2R5D, IFNAR1, CRTC2, PIK3CD, PDGFD, FGF8, FGF1, CSF1, PDGFA
	hsa-miR-503-5p	MYB, CCND2, IGF1R, GHR, EIF4E, PIK3R1, INSR, FGF2, AKT3, CCNE1, VEGFA, FGF7, CCND3, COL4A1
	hsa-miR-147a	CREB3L3, FGF14, BCL2, CDK6, FGF11, ITGA11, PDGFB, LPAR5, PDGFC, TNR, PDPK1, FGFR2, PPP2R1B
	hsa-miR-106b-5p	PHLPP2, RBL2, PDGFRA, ITGB8, PPP2CA, CREB5, MCL1, CCND1, EIF4E, PPP2R2A, F2R, FLT1, SOS1, CHRM2, EIF4E2, PDGFD, OSM, PKN2, CDKN1A, MAPK1, **JAK1**, GNB5, BCL2L11
	hsa-miR-545-3p	PRLR, PDGFB, IRS1, PDGFD, RPS6KB1
	hsa-miR-204-5p	CREB5, CCND2, FGF20, COL6A6, IL7R, CREB1, SOS1, COL5A3
	hsa-miR-425-5p	YWHAG, CREB1, SOS1, FGF9, G6PC2, IGF1, TNR, PTEN
	hsa-miR-450a-5p	CREB1
	hsa-miR-140-5p	PDGFRA, GNG12, EFNA4, FGF9, BCL2L1, CREB3L1, PKN2, GYS1, TNN
	hsa-miR-30b-3p	PHLPP2, GSK3B, COL27A1, EFNA5, FGF11, EIF4B, COL11A2, VWF, MAPK1
	hsa-miR-185-3p	PRKCA, GNG13, FGF17, CRTC2, SGK2, MAPK1
	hsa-miR-185-5p	PHLPP2, CDK2, CCND2, GNB3, IL7, ITGA1, CDK6, BRCA1, EIF4B, EIF4E2, RELA, VEGFA, PPP2R1B
	hsa-miR-431-5p	ITGA1, LAMC2
	hsa-miR-30c-1-3p	PHLPP2, GSK3B, G6PC3, FGF11, SOS1
	hsa-miR-920	IL2RB, PCK2, ITGA3, CSF1
	hsa-miR-671-5p	NFKB1, **THBS1**, PPP2R5C, BRCA1, KIT, PRKAA1, SGK2
	hsa-miR-622	ATF2, CREB5, ITGB5, COL2A1, PTEN
	hsa-miR-146b-5p	PDGFRA, NRAS, COL4A3
	hsa-miR-542-5p	IKBKB
	hsa-miR-219a-1-3p	FASLG, IGF1, IL6R
	hsa-miR-425-3p	**FN1**
FoxO signalling pathway	hsa-miR-148b-3p	SOS2, NRAS, CDKN1B, GADD45A, PIK3R3, S1PR1, SOS1, PRKAA1, IGF1, TGFB2, HOMER1, USP7, PTEN, BCL2L11
	hsa-miR-671-5p	STK4, SMAD3, PRKAA1, SGK2, MAPK10
	hsa-miR-503-5p	CCND2, IGF1R, PIK3R1, INSR, AKT3
	hsa-miR-665	STAT3, TGFBR1, SETD7, NLK, AKT3, SOD2, CDKN1A
	hsa-miR-132-3p	SMAD2, PRKAA2, SIRT1, KRAS, NLK, SOS1, TGFB2, EP300, AKT3, SOD2, PIK3CA, FOXO3, PTEN, SGK3, MAPK1, C8orf44-SGK3
	hsa-miR-212-3p	SMAD2, PRKAA2, SIRT1, KRAS, NLK, SOS1, TGFB2, EP300, AKT3, SOD2, PIK3CA, FOXO3, PTEN, SGK3, MAPK1, C8orf44-SGK3
	hsa-miR-1237-3p	TGFBR1, FBXO32, SMAD2, PRKAA2, CCND2, MAPK11, BCL6, PIK3CA, USP7
	hsa-miR-106b-5p	BRAF, RBL2, STAT3, SLC2A4, HOMER2, CCND1, SMAD4, S1PR1, MAPK8, SOS1, SOD2, CDKN1A, MAPK1, CCNG2, TGFBR2, BCL2L11
	hsa-miR-147a	RAG1, CDKN2B, NLK, FOXG1, EP300, PDPK1, MAPK10
	hsa-miR-34c-3p	SMAD2, SOS1, HOMER1, PLK4, GRB2
	hsa-miR-30c-1-3p	SMAD2, STK4, G6PC3, SOS1, SOD2, HOMER1
	hsa-miR-22-5p	PIK3CB, RAF1, IL7R, SOS1, CSNK1E, MAP2K1, SGK3
	hsa-miR-1244	FBXO25, STAT3, MAPK8
	hsa-miR-30e-5p	IRS2, FBXO32, SMAD2, SIRT1, SETD7, **CAT**, IGF1R, KRAS, PIK3CD, SKP2, MAPK8, FOXG1, SOS1, IRS1, BCL6, FOXO3, ATG12, BCL2L11
	hsa-miR-622	SIRT1, PTEN
	hsa-miR-32-5p	IRS2, BRAF, RAG1, FBXO32, PRKAA2, PIK3CB, SETD7, KLF2, NLK, PIK3R3, PIK3CA, PTEN, SGK3, BCL2L11
	hsa-miR-1205	SLC2A4, SMAD2, BNIP3, PIK3R2, PIK3R5, CCND2, IKBKB, SMAD4, GRM1, BCL6, SOD2, MAPK1, BCL2L11
	hsa-miR-425-5p	MAPK8, GRM1, SOS1, G6PC2, IGF1, SOD, PTEN
	hsa-let-7i-5p	IRS2, TGFBR1, NRAS, CCND2, IGF1R, TNFSF10, HOMER2, MAPK8, INSR, IGF1, CDKN1A, IL10
	hsa-miR-1-3p	**EGFR**, IKBKB, SOS1, IGF1
	hsa-miR-219a-1-3p	SMAD2, FASLG, IGF1
	hsa-miR-766-3p	FBXO32, PIK3R5, IKBKB, NLK, SKP2
	hsa-miR-582-3p	CDKN1B, FOXO3, PTEN, CREBBP
	hsa-miR-185-3p	MAPK14, CDK2, CCND2
	hsa-miR-326	RAF1, SMAD3, IKBKB, SGK2
	hsa-miR-30b-3p	SETD7, SOD2, MAPK1
	hsa-miR-146b-5p	BRAF, NRAS, SMAD4, SOD2
	hsa-miR-204-5p	CCND2, IL7R, SOS1
	hsa-miR-484	PIK3CD, GRM1
	hsa-miR-431-5p	IRS2
	hsa-miR-602	FOXG1
	hsa-miR-140-5p	NLK, GABARAPL1, CSNK1E
	hsa-miR-920	PCK2
	hsa-miR-542-5p	IKBKB
	hsa-miR-545-3p	STAT3, IRS1
	hsa-miR-146b-3p	RAF1
	hsa-miR-185-5p	MAPK14, CDK2, CCND2
Rap1 signalling pathway	sa-miR-106b-5p	BRAF, PDGFRA, SIPA1L3, CRK, MAGI3, TIAM1, F2RL3, RAPGEF4, PFN2, F2R, FLT1, DOCK4, PDGFD, MAPK1, RAPGEF5, GRIN2B
	hsa-let-7i-5p	ACTB, NRAS, IGF1R, FGF11, INSR, NGF, IGF1, PARD6B
	hsa-miR-148b-3p	NRAS, MRAS, PIK3R3, FLT1, KITLG, IGF1, CSF1, TEK, GRIN2B
	hsa-miR-146b-5p	BRAF, PDGFRA, NRAS, SIPA1L1, GRIN2B
	hsa-miR-431-5p	PLCB1, RAPGEF2, PRKD1, RAPGEF5
	hsa-miR-503-5p	CRK, IGF1R, RAPGEF2, PIK3R1, INSR, FGF2, AKT3, SIPA1L2, VEGFA, FGF7, PARD6B
	hsa-miR-30e-5p	MAGI2, ITGB1, EFNA3, IGF1R, FGF20, KRAS, RAPGEF4, PIK3CD, PFN2, RAPGEF2, FLT1, DOCK4, GNAQ, PDGFC, GNAI2, RASGRP3, GRIN2A, RAP1B, PDGFA, **ITGB3**
	hsa-miR-1-3p	ACTB, MAGI2, MET, ADCY1, RAP1A, **EGFR**, CALM2, PFN2, RAPGEF2, FYB, IGF1, MAGI1, RAP1B, PDGFA
	hsa-miR-1244	LPAR3, RHOA, RAPGEF2, HGF, RAP1B
	hsa-miR-582-3p	RHOA, RAPGEF2, ANGPT1
	hsa-miR-324-3p	RAP1GAP
	hsa-miR-920	RAP1GAP, GNAO1, CSF1
	hsa-miR-32-5p	BRAF, PIK3CB, RAP1A, ADCY3, PIK3R3, GNAQ, PIK3CA, RAP1B
	hsa-miR-22-5p	MAGI2, PRKCA, PIK3CB, RAP1A, RAF1, BCAR1, CNR1, MAP2K1, GRIN2B
	hsa-miR-30b-3p	PRKCI, EFNA5, VAV2, FGF11, **CDH1**, RALB, MAPK1, GRIN2B
	hsa-miR-185-3p	PRKCA, FGF17, MAPK13, MAPK1
	hsa-miR-147a	FGF14, CRKL, ADCY2, CALM1, TIAM1, FGF11, PLCG1, ITGAM, PDGFB, LPAR5, PRKCB, PDGFC, FGFR2, GRIN2B
	hsa-miR-545-3p	CRK, PFN2, PDGFB, GNAQ, PDGFD
	hsa-miR-484	RAPGEF3, LAT, CALM1, RASGRP2, F2RL3, **CDH1**, PIK3CD, PDGFD, FGF8, GRIN1, FGF1, CSF1, PDGFA
	hsa-miR-140-5p	PDGFRA, ADCY7, RALA, EFNA4, DRD2, PFN2, RAPGEF6, FGF9, FYB, GNAQ, ADCY6
	hsa-miR-326	RAF1, MAGI3, MRAS, FGF9, FGF18, FGFR2, TLN1, FGF1, CSF1, GRIN2B
	hsa-miR-185-5p	ADCY2, MAPK14, MAP2K6, SIPA1L1, CDC42, PRKCB, VEGFA, ADCY4, GRIN2B
	hsa-miR-34c-3p	MAGI3, MRAS, MAP2K6
	hsa-miR-30c-1-3p	MAP2K6, FGF11, GRIN2A, GRIN2B
	hsa-miR-766-3p	PIK3R5, RASGRP2, PRKCG, ADCY4, PLCB2, GRIN2B
	hsa-miR-1205	MET, RAPGEF1, CTNND1, PIK3R2, PIK3R5, RASGRP2, SPECC1L-ADORA2A, RAPGEF6, RASGRP3, MAPK1, PDGFRB, GRIN2B
	hsa-miR-665	RALA, FGF2, AKT3, MAGI1, **KDR**, GRIN2B
	hsa-miR-204-5p	FGF20, MAGI1, ADCY6
	hsa-miR-425-5p	CTNND1, FGF9, PRKCB, IGF1, GNAQ, GRIN2B
	hsa-miR-132-3p	CRK, ADCY3, MAGI3, KRAS, AKT3, PIK3CA, PRKD1, MAPK1, GRIN2B
	hsa-miR-212-3p	CRK, ADCY3, MAGI3, KRAS, AKT3, PIK3CA, PRKD1, MAPK1, GRIN2B
	hsa-miR-1237-3p	ACTB, SPECC1L-ADORA2A, MAPK11, GNAI2, PIK3CA, FGF1, GNAI1, GRIN2B
	hsa-miR-146b-3p	RAF1, MLLT4, LPAR5, PRKCB
	hsa-miR-671-5p	**THBS1**, RAPGEF6, KIT, ADCY4
	hsa-miR-622	PLCB1
	hsa-miR-219a-1-3p	IGF1, SIPA1L2
	hsa-miR-602	RASGRP2
	hsa-miR-877-5p	RASGRP3

## Data Availability

The datasets used or analysed during the current study available from the corresponding author on reasonable request.

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
