# Peer review of "Co-Expression Analysis of microRNAs and Proteins in Brain of Alzheimer’s Disease Patients"

_cells, 2022, doi:10.3390/cells11010163_

Round 1

Reviewer 1 Report

The authors investigate the expression correlation of microRNAs and proteins in brain of Alzheimer’s Disease patients to identify new biomarkers for AD patients. This is an interesting study.

However, there are some concerns raised:

1, Previous studies have extensively investigated mircroRNA in AD blood samples (MicroRNAs in Alzheimer’s Disease: Function and Potential Applications as Diagnostic Biomarkers, Front. Mol. Neurosci., 21 August 2020; Prediction of differentially expressed microRNAs in blood as potential biomarkers for Alzheimer’s disease by meta-analysis and adaptive boosting ensemble learning, Alzheimer's Research & Therapy volume 13, Article number: 126 (2021)). Some potential AD biomarkers have been identified. Therefore the authors should discuss whether their findings can be correlated to AD blood microRNA. The discussion on microRNA profiles in brain and blood of AD patients can provide new clues for future studies.

2, The authors use protein array to detect protein expression in AD brain samples. However, the authors should perform Western blot analysis of some key proteins to double confirm the changes of protein expression in AD patients. As fake positive results may be achieved in protein array analysis.

3, Some minor defects:

In line 81, only Mir- is shown, the microRNA number is missing.

In line 359, A aggregation is shown, APP aggregation?

Author Response

We would like to thank the reviewer for the thoughtful comments and suggestions for improving the manuscript. We have fully taken the comments on board and have extensively revised the manuscript accordingly. Please find below a detailed point-by-point response to all comments that can also be followed using the “tracked change” in the main manuscript.

  • Previous studies have extensively investigated mircroRNA in AD blood samples (MicroRNAs in Alzheimer’s Disease: Function and Potential Applications as Diagnostic Biomarkers, Front. Mol. Neurosci., 21 August 2020; Prediction of differentially expressed microRNAs in blood as potential biomarkers for Alzheimer’s disease by meta-analysis and adaptive boosting ensemble learning, Alzheimer's Research & Therapy volume 13, Article number: 126 (2021)). Some potential AD biomarkers have been identified. Therefore the authors should discuss whether their findings can be correlated to AD blood microRNA. The discussion on microRNA profiles in brain and blood of AD patients can provide new clues for future studies.

 An additional table (table 3) has been added  to the SM, showing the matched miRNAs which were differentially expressed in biofluids (blood/CSF) and brain tissue of AD patients. 14 miRNAs have been found in common and these miRNAs have the potential to monitor non-invasively any change in the brain related to the onset or the progression of AD disease. These results have been commented in the discussion of the main manuscript.

  • The authors use protein array to detect protein expression in AD brain samples. However, the authors should perform Western blot analysis of some key proteins to double confirm the changes of protein expression in AD patients. As fake positive results may be achieved in protein array analysis.

Two proteins found differentially expressed in AD patients were validated by ELISA kits in the same samples. Procedure and results are now available in SM 

  • Some minor defects:

In line 81, only Mir- is shown, the microRNA number is missing. Amended

In line 359, A aggregation is shown, APP aggregation? Amended

Reviewer 2 Report

The article coexpression analysis of microRNAs and proteins in brain of Alzheimer´s Disease patients"  analyzes the miRNAs and proteins regulated in AD. The manuscript is interesting and results are original. However there are some concerns that should be addressed:

1.- There are some minor deletions in the text. For example, in line 48 APP to amyloid-bet (A----), line 51 (fluid (CSF) biomarkers of A--- or line 112 (were calculated using the----. Please review the manuscript and fill these small gaps.

2.- Do you observe any difference in miRNAs or proteins between III-IV and V-VI groups? May be useful these biomarker to differenciate the grade of AD?

3.- miRNAs and protein results should be validated by qPCR or western blotting? This point is important at least for the most regulated biomarkers.

4.- The study of these biomarkers in brain samples make it difficult to apply in clinical practice. Are any of these miRNAs or proteins regulated in blood samples?

5.- How authors analyze RNA quality and integrity? Please indicate this information because RNA quiality is a crucial point for miRNAs studies

Author Response

We would like to thank the reviewer for the thoughtful comments and suggestions for improving the manuscript. We have fully taken the comments on board and have extensively revised the manuscript accordingly. Please find below a detailed point-by-point response to all comments that can also be followed using the “tracked change” in the main manuscript.

  • There are some minor deletions in the text. For example, in line 48 APP to amyloid-bet (A----), line 51 (fluid (CSF) biomarkers of A--- or line 112 (were calculated using the----. Please review the manuscript and fill these small gaps.

Checked and corrected

  • Do you observe any difference in miRNAs or proteins between III-IV and V-VI groups? May be useful these biomarker to differentiate the grade of AD?

Two additional tables were added to the  SM (table 1 and 2) showing the differences of either miRNAs or proteins between Braak II-IV and Braak V-VI stages

  • miRNAs and protein results should be validated by qPCR or western blotting? This point is important at least for the most regulated biomarkers.

Two proteins found differentially expressed in AD patients were validated by ELISA kits in the same samples. Procedure and results are now available in SM 

  • The study of these biomarkers in brain samples make it difficult to apply in clinical practice. Are any of these miRNAs or proteins regulated in blood samples?

An additional table (table 3) has been added to SM showing the matched miRNAs which were differentially expressed in biofluids (blood/CSF) and brain tissue. 14 miRNAs have been found in common and these miRNAs have the potential to monitor non-invasively any change in the brain or the progression of AD disease. These results have been commented in the discussion of the main manuscript.

  • How authors analyze RNA quality and integrity? Please indicate this information because RNA quiality is a crucial point for miRNAs studies

“An Agilent 2100 Bioanalyzer (Santa Clara, CA, United States) was used to detect the size distribution of total RNA, as well as determine the quality of the RNA”.

This info has been added in the method section

Round 2

Reviewer 1 Report

Can be accepted for publication.

Reviewer 2 Report

The manuscript has been improved and in my opinion this version can be published in Cells Journal